# Zinc Finger Proteins in the War on Gastric Cancer: Molecular Mechanism and Clinical Potential

**DOI:** 10.3390/cells12091314

**Published:** 2023-05-04

**Authors:** Shujie Liu, Xingzhu Liu, Xin Lin, Hongping Chen

**Affiliations:** 1Department of Histology and Embryology, Medical College, Nanchang University, Nanchang 330006, China; jp4217119023@qmul.ac.uk (S.L.); jp4217119083@qmul.ac.uk (X.L.); jp4217119011@qmul.ac.uk (X.L.); 2Medical Department, Queen Mary School, Nanchang University, Nanchang 330006, China

**Keywords:** zinc finger proteins (ZFPs), gastric cancer (GC), proliferation, biological function, prognosis, therapeutic intervention

## Abstract

According to the 2020 global cancer data released by the World Cancer Research Fund (WCRF) International, gastric cancer (GC) is the fifth most common cancer worldwide, with yearly increasing incidence and the second-highest fatality rate in malignancies. Despite the contemporary ambiguous molecular mechanisms in GC pathogenesis, numerous in-depth studies have demonstrated that zinc finger proteins (ZFPs) are essential for the development and progression of GC. ZFPs are a class of transcription factors with finger-like domains that bind to Zn^2+^ extensively and participate in gene replication, cell differentiation and tumor development. In this review, we briefly outline the roles, molecular mechanisms and the latest advances in ZFPs in GC, including eight principal aspects, such as cell proliferation, epithelial–mesenchymal transition (EMT), invasion and metastasis, inflammation and immune infiltration, apoptosis, cell cycle, DNA methylation, cancer stem cells (CSCs) and drug resistance. Intriguingly, the myeloid zinc finger 1 (MZF1) possesses reversely dual roles in GC by promoting tumor proliferation or impeding cancer progression via apoptosis. Therefore, a thorough understanding of the molecular mechanism of ZFPs on GC progression will pave the solid way for screening the potentially effective diagnostic indicators, prognostic biomarkers and therapeutic targets of GC.

## 1. Introduction

Gastric cancer (GC) is one of the most frequently diagnosed gastrointestinal malignancies worldwide with insidious onset and poor prognosis, and among which stomach adenocarcinoma (STADs) is the most common [1,2]. With its incidence and mortality rates, respectively, ranking fourth and second globally, GC is now the most prevalent disease in East Asia, East Europe and South America [3,4,5]. According to research, several key risk factors contribute to GC, including geographical environment, dietary and lifestyle patterns, smoking and *Helicobacter pylori* (*Hp*) infection, etc. [6].

Comprehensive treatment based on surgical resection is still the primary choice for the clinical treatment of GC; however, the eventual postoperative mortality and survival rates remain unfavorable [7]. For instance, the 5-year survival rate of patients after radical surgery is only between 20% and 50% [8]. The therapeutic selection and prognosis of GC can be evaluated in advance via the TNM (tumor, node, metastasis) staging system, which mainly determines the histological grade based on tumor size, lymph node location and distant metastasis [9,10,11]. Generally, stage I has a better prognosis than stages II, III and IV. According to the most widely applied Lauren classification, GC can be histologically distinguished into intestinal and diffuse types [12], whereas a molecular categorization system has currently been suggested, comprising microsatellite unstable, Epstein–Barr virus positive, genomically stable and chromosomally unstable, which provides a reliable foundation for molecule-tailored personalized therapeutics for GC [13].

As canonical transcription factors, zinc finger (ZNF) proteins play precisely defined regulatory roles in physiological processes, such as transcription translation, cell differentiation and embryonic growth, by binding specific nucleotide sequences upstream of genes [14]. Recent evidence reveals that the zinc finger protein (ZFP) family is of great importance in the molecular regulation of the genesis and propagation of human malignant tumors involving the colon, breast, liver, prostate and gastric carcinomas [15,16,17,18,19]. In recent decades, the association between ZFPs and GC has attracted much attention, especially in cell proliferation, epithelial–mesenchymal transition (EMT), invasion and metastasis, inflammation and immune infiltration, apoptosis, cell cycle, DNA methylation, cancer stem cells (CSCs), drug resistance and so on (Figure 1). This review provides a detailed summary of the relationship mentioned above and explores the underlying molecules and signaling pathways, including hedgehog (Hh), PI3K/Akt/mTOR and Wnt/β-Catenin signaling pathways. Our review hopes to find full-scale ZFP mechanisms critical for GC, explore the possibility of utilizing ZFPs for a prognostic pattern for diagnosis and provide inspiring clues and directions for GC-targeted therapy.

## 2. Zinc Finger Proteins (ZFPs)

ZFPs are a family of proteins with short self-folding finger-like domains that share the common feature of binding a zinc ion (Zn^2+^) to stabilize the structure [20]. It is the largest family of transcripts, accounting for more than 2% of the sequence in the human genome [21]. Initially discovered in Xenopus oocytes in 1983 by Miller to describe the predicted finger-like profile of transcription factor IIIA (TFIIIA), this family has now been expanded with continuously discovered paralogous proteins [22]. Based on their conserved domains, ZFPs can be mainly divided into three categories, which are Cys2His2 (C2H2)-type, Cys4 (C4)-type and Cys6 (C6)-type [23]. Among these, the C2H2-type zinc finger is the most widely distributed of the entire zinc finger motif classes [24]. According to different spatial structures formed by the zinc finger Cys and His residues surrounding the zinc ion, ZFPs could be classified into eight different folding groups, namely C2H2, Treble clef, Zn2/Cys6, Gag knuckle, Zinc ribbon, Metallothionein, TAZ2-domain-like and Zinc-binding loops [25].

Due to their characteristic structure, ZFPs can specifically bind to target DNA or RNA and in turn perform gene regulatory functions [26]. The zinc finger structure is dominated by three peptide segments that self-fold into a “finger” shape, consisting of an alpha-helix at the C-terminal and two antiparallel beta-folds at the N-terminal [27]. The conserved repeat is cysteine (Cys)-n2-4-Cys-N12-14-histidine (His)-N3-His; more specifically, two Cys sulfuryl groups and two His imidazolyl groups in this repetition domain are coordinated with Zn^2+^ and stabilized via hydrophobic interaction [28]. Furthermore, ZFPs encompass other functional domains, such as Krüppel-associated box (KRAB), sre-ZBPbu, CTfin51, AW1 and Number 18 cDNA (SCAN), as well as the broad complex, tramtrack and bric-a-brac/poxvirus and zinc finger (BTB/POZ) domains [29]. These functional domains may control DNA binding, subcellular localization and gene expression via regulating selective binding proteins [30,31,32].

## 3. Biological Functions of ZFPs in GC

Since numerous research has discovered a substantial connection between ZFPs and GC, a systematic review is urgently needed in this field. GC is still one of the most common malignancies worldwide due to its high morbidity and mortality; so, it is a worthy endeavor to make a summary of the research advances in ZFPs and GC, especially regarding eight main aspects, comprising cell proliferation, EMT, invasion and metastasis, inflammation and immune infiltration, apoptosis, cell cycle, DNA methylation, CSCs and drug resistance (Figure 1). In this section, we will make a summary of the regulatory role of ZFPs in these biological processes of GC and highlight the molecular mechanisms involved (Table 1).

### 3.1. ZFPs Regulate Cell Proliferation

Normally, cell proliferation is the basis for the growth, reproduction and inheritance of an organism [116]. However, rapidly unrestricted or uncontrollable cell proliferation will facilitate the malignancy’s transformation [117,118]. Previous studies have testified that diversified C2H2-type ZFPs exert regulatory effects in the cell proliferation of GC, which may serve as potential molecular therapeutic targets for GC.

In GC, a large number of highly expressed C2H2-type ZFPs promote cell proliferation, specifically ZNF852, ZNF521, ZNF460, ZNF280B, ZNF143, zinc finger protein X-linked (ZFX), DAZ-interacting zinc finger protein 1 (DZIP1), E3 ubiquitin ligase ring finger protein 114 (RNF114) and pleomorphic adenoma gene like-2 (PLAGL2). Ke et al. found that the proliferation of GC cells could be suppressed when using a CRISPR/CAS803 system to knock out ZNF852 due to the downregulated epidermal growth factor receptor (EGFR) expression by ZNF852 deficiency [33]. A colony formation assay indicated that ZNF521 is substantially expressed in GC cells. ZNF521 accelerates hematopoietic development and leukoplakia by raising c-Myc, and inhibits red blood cell differentiation by binding to GATA-binding protein 1. Yet, the precise mechanism of ZNF521 in stomach carcinogenesis remains unclear [36]. ZNF460 may bind to the apolipoprotein C1 (APOC1) promoter to increase APOC1 expression, boosting the development of GC [39]. Zhai et al. observed in a xenograft study that the overexpression of ZNF280B can promote GC aggressiveness in vivo, but the specific molecular mechanism requires further investigation [41]. As an oncogenic protein, ZNF143 decreased the reactive oxygen species (ROS) level, inhibited apoptosis and promoted proliferation in GC cells via the ROS/p53 axis [43]. ZFX, a zinc finger transcription factor encoded on the X chromosome, was significantly upregulated in GC. Wu et al. pointed out for the first time that ZFX has potent tumorigenicity by upregulating the extracellular-signal-regulated kinase/mitogen-activated protein kinase (ERK-MAPK) pathway [51]. DZIP1 is upregulated in cancer-associated fibroblasts (CAFs) and malignant epithelial cells [65]. RNF114 is highly expressed and participates in GC creation by modulating early growth response 1 (EGR1) transcription [66]. Studies have shown that PLAGL2 can activate the transcription of the deubiquitinating enzyme USP37 that deubiquitinates and stabilizes Snail family transcriptional repressor 1 (Snail1), finally fostering Snail1-mediated cell proliferation [62]. Interestingly, both PLAGL2 and Snail1 belong to ZFPs, having a synergistic effect on partial ZFPs in GC.

Multiple C2H2-type ZFPs can also be suppressed, thus promoting GC development, specifically ZNF545, ZNF479, ZNF471, ZNF331 and Krüppel-like factors (KLF4 and KLF6). Wang et al. found that restoring ZNF545 expression could inhibit GC cell proliferation, validating the fact that ZNF545 can act as a tumor suppressor to repress the transcription of ribosomal RNA (rRNA) and recruit corepressor and heterochromatin protein 1β [35]. As an oncogenic protein, ZNF479 knockdown was clarified to impede GC progression by regulating the β-catenin/c-Myc signaling pathway [37]. ZNF471 could directly bind to or recruit KRAB-associated protein 1 (KAP1) to the promoter of plastin 3 (PLS3) and transcription factor AP-2 alpha (TFAP2A), and inhibit their expression at the transcriptional level, thus controlling gastric carcinogenesis and development [38]. As a tumor suppressor, ZNF331 inhibits GC progression by downregulating genes that promote cell growth, such as DDX5, DSTN, EIF5A, GARS, UQCRFS1, STAM and SET [40]. In GC, the overexpression of KLF4 can significantly suppress the expression of β-catenin and matrix metallopeptidase 2 (MMP2), restore epithelial cell marker (E-cadherin) expression and significantly inhibit the colony formation of GC cells [68]. Another tumor suppressor KLF6 can inhibit GC via the regulation of the expression of c-Myc and a cyclin-dependent kinase inhibitor named p21 [70]. 

Several ZFPs with special structures can also participate in the proliferation of GC cells, such as zinc finger CCCH-type containing 15 (ZC3H15), speckle-type POZ protein (SPOP), microrchidia family CW-type zinc finger 2 (MORC2) and zinc and ring finger 3 (ZNRF3). ZC3H15 is a highly functional and evolutionarily conserved protein [119]. Considering that FBXW7 is a major contributor to c-Myc degradation, ZC3H15 regulates c-Myc protein integrity by reducing the FBXW7 transcription, thereby promoting GC cell proliferation [54]. SPOP is an E3 ubiquitin ligase adaptor and constitutes an internal BTB/POZ domain, an N-terminal math domain and a back domain [120]. Zeng et al. confirmed that SPOP can prevent the proliferation of tumor cells in GC, and the specific mechanism was that SPOP can inhibit the Hh/GLI2 signaling pathway and accelerate GLI2 degradation to inhibit GC tumorigenesis [59]. Human MORC2 is a member of the MORC protein family containing a CW-type zinc finger domain [121]. The overexpression of MORC2 modifies the TE-III domain of CCAAT/enhancer-binding protein α (C/EBP-α) via sumoylation, which promotes the sumoylation and subsequent degradation of C/EBP-α protein, affects its protein stability and then leads to cell proliferation and tumorigenesis [67]. ZNRF3, belonging to the E3 ubiquitin ligases family, acts via the Wnt/β-Catenin/TCF pathway to inhibit cancer cell survival and growth in GC [113].

### 3.2. ZFPs Regulate EMT, Invasion and Metastasis

A significant process in the invasion and metastasis of malignant tumors is EMT, in which epithelial cells undergo a loss of polarity and cell–cell adhesions and change into motile mesenchymal-like cells [122]. Both E-cadherin and mesenchymal cell markers (e.g., N-cadherin and Vimentin) are crucial for monitoring the dynamic process of EMT [123]. Apart from preserving the shape, polarity and integrity of epithelial cells, E-cadherin is in charge of the adhesion and connection between epithelial cells [124]. The force of adhesion among epithelial cells is decreased and the incidence of EMT is enhanced when the E-cadherin expression level is relatively low [125]. Tumor invasion and metastasis are multi-stage processes whereby malignant tumor cells detach from their primary sites, reach distant sites and colonize there [126]. This process depends on the degradation of the extracellular matrix (ECM) and the loss of the structural stability of the basement membrane. Matrix metalloproteinases (MMPs), a type of proteolytic enzyme, can complete the breakdown of ECM and be suppressed by metal matrix protease inhibitors (TIMPs) [127]. Of note, the classification of ZFPs in their various roles in three distinct dimensions, i.e., EMT (Figure 2A), invasion (Figure 2B) and metastasis (Figure 2C), and an overview of metastatic dissemination in GC progression are depicted in Figure 2.

Amounts of C2H2-type ZNFs with a high expression which can trigger EMT, invasion and metastasis were found in GC, particularly ZNF521, ZNF460, ZNF143, zinc finger protein 139 (ZNF139), zinc finger and BTB domain-containing 20 (ZBTB20), Snail, DZIP1, Slug, RNF114, PLAGL2, GLI family zinc finger 1 (GLI1), Krüppel-like factor 8 (KLF8), CCCTC-binding factor (CTCF) and the E-box-binding protein ZEBs (ZEB1 and ZEB2), as illustrated in Figure 2. The emerging evidence has proved that miRNA-204-5p, a crucial member of the miRNA family, controlled the progression of GC [128,129]. Moreover, Huan et al. uncovered that suppressed miRNA-204-5p can enable GC cells to invade and metastasize via the upregulation of ZNF521 [36]. Other ZFPs can also be affected by miRNA. For instance, miR-BART12 can repress and degrade Snail, thus attenuating the migration and EMT process prompted by Snail to exert antineoplasm effects [63]. In addition, ZNF460 integrates with APOC1 to promote APOC1 transcription to enhance EMT, thus accelerating GC progression in a recent in vivo model [39]. In terms of ZNF143, Wei et al. indicated that a low expression of ZNF143 can inhibit distant metastasis in mice, and a high ectopic expression of ZNF143 can promote GC cell invasion in vitro. Additionally, they demonstrated that ZNF143 can attenuate E-cadherin expression while enhancing N-cadherin expression, thus accelerating the EMT process via the PI3K/Akt signaling pathway [42]. Curiously, they also discovered that ZNF143 can enhance the EMT process by promoting other ZFPs (Slug and Snail) [42]. Notably, many other cross-modulations between ZFPs also exist in the EMT and aggressiveness of GC cells. The transcriptional activator PLAGL2 activates the ubiquitin-specific peptidase 13 (USP13), which deubiquitinates and maintains Snail to influence the metastatic power of GC cells [64]. Likewise, Slug (Snail2) can work synergistically with Snail and SIP1 to repress E-cadherin expression to prompt the EMT process of the diffuse GC [61]. Furthermore, Li et al. investigated the role of ZNF139 in modulating the invasion and colonization activity of GC cells via siRNA technology. Mechanistically, ZNF139 can interfere with the balance of MMPs-TIMP via upregulating MMP-2 and MMP-9 and downregulating TIMP-1 to further induce the migration and metastasis of GC cells [46]. In addition, ZBTB20 facilitates the cell invasion and metastasis of GC via blocking IκBα or inducing NF-κB activation [55]. Another oncogenic protein DZIP1 also participates in altering the immunosuppressive microenvironment homeostasis and promotes EMT via activated CAF, which is the principal participator in ECM stiffness and degradation [65]. In addition, RNF114 is another pivotal biomarker in GC progression which can promote the migration and growth of GC cells by activating ECR1 ubiquitylation and degradation [66]. Moreover, Liang et al. discovered that GLI1 regulated the EMT process of GC cells via transforming growth factor β1 (TGF-β1) [76]. Noteworthily, KLF8 is another downstream transcription factor of TGF-β1. Zhang et al. uncovered that silenced KLF8 can reverse the reduction of E-cadherin, which illustrates the KLF8 participation in TGF-β1-induced EMT [82]. A migration assay discovered that CTCF can target ECM-related genes, namely COL1A1 and COLA31 in vitro, and repressed CTCF, COLIA1 and COLA31 could impede the invasion and metastasis of GC cells [79]. Critically, the zinc finger E-box-binding homeobox (ZEB) family is composed of two key EMT-related proteins, ZEB1 and ZEB2. The ZEB family can suppress E-cadherin at the transcriptional level, induce EMT in epithelial cells and enhance the invasion and motility of GC cells. Lu et al. demonstrated that aryl sulfonamide indisulam curbs GC invasion via the downregulation of ZEB1 by the ubiquitin-proteasome system (UPS), a complicated intracellular degradation system responsible for regulating protein activity and stability [108]. ZEB2-antisense RNA1 (ZEB2-AS1) can induce the EMT process by upregulating MMP-2 and MMP-9 in AGS cells, thus prompting the invasion and metastasis of GC cells [109]. Nonetheless, there is a dearth of research on ZEB2 and GC at the molecular level now, and further study is still required to determine whether ZEB2 is precisely linked to GC.

Certain C2H2-type ZNFs, notably ZNF471, ZNF331, ZNF24, KLF4, Krüppel-like factor 9 (KLF9), ZNF382 and ring finger protein 180 (RNF180), can be suppressed to inhibit EMT, migration and metastasis, thus halting gastric tumorigenesis (Figure 2). The ectopic expression of ZNF471 was indicated to suppress the invasion and metastasis of GC cell lines in mice models [38]. In GC, ZNF331 can also curb tumor cell invasion by downregulating DSTN and ACTR3 [40]. Additionally, ZNF24 is downregulated by microRNA-940, which enhances the migration and metastasis of GC cells [49]. Interestingly, the KLF family also participates in the metastasis of GC, involving KLF4 and KLF9. The overexpression of KLF4 can reverse the loss of E-cadherin expression and impede MMP2 expression in GC cells [68]. KLF9 can prevent GC metastasis by suppressing the transcriptional expression of MMP28, which is distinct from other members of C2H2-type ZNFs [72]. Mechanistically, KLP9 participates in the TPTEP1/miR-548d-3p/KLF9/PER1 axis to modulate GC progression [71]. ZNF382 is another tumor suppressor which upregulates the expression of E-cadherin to halt the EMT process in GC cells via NOTCH signaling [111]. RNF180 can induce the ubiquitination and degradation of DNA methyltransferase 3α (DNMT3A), thus restoring ADAMTS9 expression to impede metastasis in GC cells [85]. Wu et al. also indicated that RNF180 can ubiquitinate RhoC protein to trigger its breakdown to suppress the phosphorylation of STAT3 to hinder GC progression [83].

Some ZNFs with unique structures can also participate in inducing EMT and the motility and invasion of GC cells, namely highly expressed ZNFs including ZC3H15, Twist family bHLH transcription factor 1 (TWIST1) and MORC2 (Figure 2). ZC3H15 functions by concentrating on the FBXW7/c-Myc pathway to accelerate the progression of GC [54]. TWIST1 which belongs to the bHLH family has an association with the EMT process [130]. Romero et al. unveiled that polymorphism in TWIST1 in GC can affect EMT markers and further in vitro and in vivo experiments are still required to be performed [58]. MORC2 contains a CW-type zinc finger region and the aggressive phenotypes of GC patients were linked to elevated MORC2 expression, as confirmed by Liu et al. [67]. Conversely, the suppressed ZNFs in GC such as SPOP can impede GC progression, and in vitro tests suggested that overexpressed SPOP can block the invasion and colony formation of tumor cells by suppressing the Hh/GLI2 signaling pathway [59].

### 3.3. ZFPs Regulate Inflammation and Immune Infiltration

Tumor cells, inflammatory cells and adjacent cellular stroma in chronic and recurrent inflammation can create an inflammatory tumor microenvironment (TME) conducive to tumor progression [131]. The primary cause of sporadic GC is *H. pylori* infection, and its colonization in GC epithelial cells can result in a series of inflammatory premalignant events [132]. A recent risk model toward inflammation in GC was constructed, which demonstrated that the inflammation-related genes were highly correlated with immune infiltration [133]. Thus, we need to elucidate how the immunological microenvironment contributes to the development of chronic inflammation in GC. Few studies have shown that certain highly expressed ZNFs can aggravate the malignancy of GC via the modulation of inflammation and immunosuppression, represented by zinc fingers and homeoboxes 2 (ZHX2), ZFP64 and DZIPI in GC. There is a positive relationship between ZHX2 and immune infiltration cells, involving dendritic cells, B cells, macrophages and especially T helper cells in GC, thus enabling the promotion of the spread of invasive GC [52]. Additionally, the overexpression of ZFP64 can increase CD8^+^ T cells and several immunosuppressive cytokines such as CXCL10, IL-1α and CSF1, while decreasing IFN-γ and IL-2 to trigger GC progression [53]. Elevated DZIPI can promote GC progression via CD163 which is used to label anti-inflammatory M2 tumor-related macrophages to hinder T-cell infiltration [65]. Additionally, TNFα-induced protein 3 (TNFAIP3, A20) has been well studied in regard to the immune response and inflammatory process in tumors, and it is postulated that TNFAIP3 may have immunomodulatory potential in GC; further research is still required to verify this [80]. In conclusion, it is urgent to focus more on this aspect to further facilitate immunopharmacology with precise mapping and powerful targeting therapy.

### 3.4. ZFPs Regulate Apoptosis

Apoptosis, also known as programmed cell death, is tightly regulated by genes and proteins. Its dysfunction may result in uncontrolled growth and tumor formation [134]. At present, apoptosis has been identified as the primary therapeutic strategy of numerous anticancer medications; henceforth, the potential mechanism of apoptosis between ZFPs and GC is a new research hotspot [135,136].

Many C2H2-type ZFPs are closely associated with apoptosis, such as ZNF143, GLI1, ZNF139 and ZFX, and all of them are upregulated. A recent study showed that ZNF143 is associated with apoptosis triggered by ROS-induced oxidative damage in vivo [43]. ZNF143 can decrease ROS levels and inhibit apoptosis in GC cells. Further studies showed that p53 transfection can reverse the anti-apoptotic action of ZNF143, while the p53-specific inhibitor pifithrin-α reduced the apoptotic influence of ZNF143, which might be based on the inhibition of ROS production in GC cells by P53 protein, thereby reducing the apoptotic effect of tumor cells. In addition to ROS, another crucial factor involved in oxidative stress NADPH Oxidase 4 (NOX4) is also related with ZFPs such as GLI1 in GC. GLI1 knockdown can reverse the effects of NOX4 overexpression, demonstrating that GLI1 is an indispensable effector of NOX4 in mediating cell apoptosis [137]. Moreover, GLI1 was significantly decreased in the NOX4 knockdown group, accompanied by a reduction in apoptotic proteins such as Bcl2, Bax and cleaved PARP, confirming that NOX4 might stimulate GLI1 and promote apoptotic protein expression to facilitate apoptosis. Moreover, itraconazole, an effective therapeutic drug based on the GLI1 regulation of downstream targets Bax and PARP, also effectively demonstrated the close relationship between GLI1 and apoptosis [138]. Li et al. showed that ZNF139 is negatively correlated with the GC cell apoptotic index (*r* = −0.686; *p* < 0.01) and the underlying molecular mechanism was that ZNF139 can attenuate apoptosis in GC tissues by inhibiting Caspase-3 and Bax while promoting Bcl-2 expression [47]. Thereafter, the experiment further confirmed that ZNF139 can promote the apoptosis resistance of GC by regulating some apoptosis-related genes, such as survivin, X-IAP, Caspase-3, Bcl-2 and Bax [16]. In GC, the reduction in ZFX can increase the number of apoptotic cells, evidenced by two critical apoptotic factors, increased Bax and decreased Bcl-2. ERK-MAPK signaling pathway activation via ZFX can suppress cell death and promote tumor development [51].

Ubiquitination is one of the most common post-translational modifications in proteomics, which covalently binds ubiquitin molecules to target proteins and regulates metabolic reprogramming in cancer [139]. In GC, multiple ZFPs regulate apoptosis via ubiquitination machinery, such as TNFAIP3, Ring finger protein 43 (RNF43), MPS-I, ZNRF3 and SPOP. The tumor necrosis factor (TNF)-related apoptosis-inducing ligand (TRAIL) can trigger apoptosis by inducing the formation of the death-inducing signaling complex (DISC) [140]. In GC, the C2H2-type TNFAIP3 can interact with RIP1 and DR4, and a low expression of TNFAIP3 can prevent the polyubiquitination of RIP1, promote the cleavage of caspase-8 and then inhibit DISC formation, eventually suppressing apoptosis [81]. RNF43 is a member of the ring domain E3 ubiquitin ligase family and plays a crucial role in GC development [96,106]. RNF43 was found to inhibit cell proliferation and promote apoptosis, which was further confirmed by the positive correlation between RNF43, p53 and cleaved-caspase3, and the negative correlation between Ki67 and Lgr5 [93]. Metallopanstimulin-1 (MPS-1) is a zinc-finger-domain-containing effective ribosomal protein RPS27 that is notably expressed in human GC [104]. The removal of MPS-1 diminishes the kinase activity of p65 and NF-κB, therefore limiting the expression of the growth arrest DNA damage inducible gene 45β (GADD45β), an immediate NF-κB target. However, a low expression of GADD45β can induce apoptosis by promoting c-Jun kinase (JNK) phosphorylation. The above research revealed the important role of the MPS-1/NF-κB signaling pathway in the apoptosis of GC induced via the knockdown of MPS-1 [105]. ZNRF3, an E3 ubiquitin ligase which is silenced or mutated in GC tissues, can significantly increase Lgr5 of the Wnt pathway and GLI1 of the Hh pathway, promote Wnt and Hh signaling and inhibit apoptosis [112]. SPOP is an E3 ubiquitin ligase adaptor that can inhibit oncogenic signaling [60]. When SPOP is elevated, apoptotic proteins related to the Hh/GLI pathway, such as Caspase-3 and PARP, are increased. When SPOP is knocked down, the tumor suppressor Phosphatase and Tensin Homolog (PTEN) and the cyclin-dependent kinase inhibitors (CDKIs) p16, p21 and p27 are reduced, while cyclin B1 and proliferating cell nuclear antigen (PCNA) are increased. Meanwhile, this study argues that SPOP may directly be associated with GLI2 within the cytoplasm to form a complex that is then ubiquitinated and degraded, thus blocking the effect of GLI2 on target gene activation and affecting the apoptosis of GC cells [59].

### 3.5. ZFPs Regulate the Cell Cycle

Under normal circumstances, cell-cycle-protein-dependent kinases (CDKs) and CDKIs precisely orchestrate the cell cycle, and CDKIs prevent CDKs from functioning [141]. Carcinoma cell cycle disruption promotes unrestrained growth and spread [142,143]. Five G2/M checkpoint-related genes, including CCNF, MAPK14, MARCKS, CHAF1A and INSENP, have been recently linked to the clinical outcome of GC patients using bioinformatics techniques [144]. More strikingly, emerging evidence suggests that ZFPs can modulate the cell cycle to regulate the progression of GC. 

In GC, the upregulation of some ZNFs can increase the malignancy of GC via cell-cycle-related regulation such as ZNF139, ZFX, MORC2, GLI1, ZNF460, Krüppel-like factor 12 (KLF12) and Ring finger protein 2 (RNF2). Fan discovered that G0/G1 cells were greatly elevated but G2/M cells were dramatically decreased when suppressing ZNF139, indicating that ZNF139 can intervene in the cell cycle to block the apoptosis of GC cells [44]. The ZFX knockdown could induce cell cycle arrest in the G1/S transition, while cyclin E1 and cyclin A2 were remarkedly downregulated [51]. MORC2 can trigger C/EBPα-facilitated C2C12 cell cycle G1/S transition to mediate poor cell differentiation in GC cells [67]. Moreover, GLI1 downregulation can disrupt the G1/S transition and impede cell growth via p21(Waf1/Cip1) modulation [145]. Additionally, An and Liu recently unearthed that ZNF460 downregulation can increase G1/S cells and decrease G2 phase cells, suggesting that ZNF460 can promote GC growth via cell cycle regulation [39]. Furthermore, KLF12 may be the putative target gene of miR-137 [102] and miR-876-3p [103], which can facilitate cell cycle arrest in the G0/G1 phase. In addition, RNF2 downregulation can reduce GC cells in the G2/M phase while enhancing the G1 phase to impede cell proliferation by upregulating CDKIs p21 (Waf1/Cip1) and p27(Kip1) [100]. Conversely, KLF6 and ZIC family member 1 (ZIC1) are two C2H2-type ZNFs that are suppressed to inhibit GC development. An overexpression of KLF6 can cause G1/S arrest to suppress the proliferation of AGS cells via the transcriptional modulation of p21 and c-Myc [70]. ZIC1 also serves as a tumor suppressor in GC to mediate the G1/S checkpoint by regulating p21, p27 and cyclin D1 via sonic hedgehog (Shh) signaling [115].

### 3.6. ZFPs Regulate DNA Methylation

DNA methylation is the most dominant form of epigenetic inheritance, characterized by unchanged nucleic acid sequence and heritable gene expression [146]. Gene silencing and transcription are repressed when the DNA of CPGs within the regulatory region is methylated [147]. Studies have found that GC is often accompanied by several special forms of ZFP DNA methylation, such as ZNF545, ZNF471, ZNF331, RNF180 and ZIC1. Strikingly, these ZFPs all belong to the C2H2-type, both function as tumor suppressors and all are downregulated in GC. Wang et al. found that the detection rate of ZNF545 methylation was 51.9% in GC tissues and 27.0% in paracancer tissues (*p* = 0.001), while no ZNF545 methylation was detected in 20 normal gastric mucosa tissues [35]. This might be the result of the hypermethylation of the ZNF545 promoter in GC, which in turn inhibited rRNA transcription. Recent research identified greater ZNF471 promoter methylation in primary GC in comparison with neighboring regular tissues (*p* < 0.001) [38]. Mechanistically, ZNF471 directly binds to or recruits KAP1 to the promoters of strongly oncogenic TFAP2A and PLS3, and then transcriptionally inhibits their expression to exert antitumor effects. ZNF331 was silenced or downregulated in 71% of GC cell lines resulting from the hypermethylation of its promoter [40]. The methylation of RNF180 contributes to decreased RNF180 expression, which is linked to the incidence and progression of GC [84]. Moreover, in GC, we also observed that ZIC1 expression was downregulated, accompanied by the hypermethylation of the ZIC1 promoter [101]. Nonetheless, the specific mechanisms of action of these three proteins still need further research and exploration.

### 3.7. ZFPs Regulate Cancer Stem Cells

Under appropriate circumstances, stem cells may self-renew or differentiate, giving rise to daughter cells with precise genotypes and phenotypes, as well as progenitor cells with particular differentiated objectives [148]. Unfortunately, extended living durations raise the risk of genetic abnormalities, permitting them to escape apoptosis and thus leading to tumor formation [149]. Recent studies have found a close relationship between GC and CSCs, especially GLI1, GLI2, ZFP64 and ZNF852, which both belong to the C2H2-type. The nuclear transcription factors GLI1 and GLI2 are essential molecules in the Shh signaling pathway, which have been implicated in sustaining CSC properties in GC [150]. Studies have revealed that GLI1 not only participates in the tumorigenesis of GC, but also upregulates CSC surface markers such as CD44, Lysine-specific demethylase 1 (LSD1) and SRY-Box Transcription Factor 9 (Sox9) [151,152], whereas GLI2 fosters the expression of CSC-related genes, such as CD44, Nanog homeobox (Nanog) and octamer-binding transcription factor 4 (Oct4), by inducing platelet-derived growth factor receptor beta (PDGFRB) [153,154]. ZFP64 directly binds to the Galectin-1 (Gal-1) promoter to further enhance Gal-1 transcription and induce an embryonic cell-like phenotype in GC [53]. Moreover, when the ZNF852 is knocked down using the CRISPR/cas9 system in GC, its ability to reproduce is inhibited, along with the decreased expression of Nanog, SRY-box 2 (Sox2) and Oct4, meaning that ZNF852 preserves the self-renewal and tumor stem cell properties of GC [33]. However, it is not clear whether ZNF852 directly enhances the transcription of Nanog, Sox2 and Oct4, or whether Sox2 and Oct4 upregulate ZNF852 in the line with positive feedback.

### 3.8. ZFPs Regulate Drug Resistance

Drug resistance leads to metastasis or recurrence in advanced cancer, which is one of the major obstacles to successful cancer treatment [155]. It develops primarily due to the acquisition of new gene mutations, and the constant evolution and adaptation of tumors, resulting in the removal or alteration of drug target molecules, bypass activation and so on [156]. The heterogeneity of cancer drug resistance remains a great challenge for treatment, and the fundamental molecular mechanisms among ZFPs, drug resistance and GC are beneficial for the discovery of new targeted therapies [157].

Right now, findings on GC prove that some C2H2-type ZFPs, for example, ZNF852, ZNF139, ZFP64, GLIS family zinc finger 2 (GLIS2), GLI1, GLI2 and ZEB2, and their expressions are all upregulated. ZNF852 deficiency can enhance oxaliplatin-induced GC cell death, suggesting that ZNF852 increases Sox2, Oct4 and Nanog transcription, which may boost carcinogenesis and enhance drug resistance [33]. ZNF139 can promote annexin A2 (ANXA2) and fascin expression and decline pyridoxal kinase (PDXK) expression, thus promoting the drug resistance of GC [45]. However, a recent study clarified that ZNF139 can repress miR-185 to increase the multi-drug resistance of GC cells [48]. Studies have determined via the RNA sequencing of samples from patients sensitive or resistant to nab-paclitaxel that a high expression of ZFP64 promotes GC progression and reduces treatment efficacy [53]. The underlying mechanism is that ZFP64 can directly bind to the Gal-1 promoter and promote Gal-1 transcription, thereby provoking a stem-like phenotype of GC and an immunosuppressive microenvironment. There are studies based on co-expression network analysis showing that GLIS2 redundancy contributes to GC chemoresistance and poor prognosis [74]. Additionally, another study proved that the low expression of GLIS2 may be significantly associated with radiosensitivity in gastric GC patients, but the specific mechanism needs further exploration [73,75]. In a bioinformatics analysis, 600 GLI1 co-expressed genes were identified, and GLI1 was found to be greatly enriched in the Hh signaling pathway and PI3K/Akt pathway, which were closely related to chemoresistance, but the specific mechanism needs further exploration [78]. GLI1 warrants further investigation in HER2-targeted therapy-resistant GC, and the underlying mechanism is that HER2 may manage GLI1 via the Akt-mTOR-p70S6K pathway, promoting GC development [77]. Beiqin Yu et al. discovered that GLI2 expression was elevated in GC cells treated with fluorouracil (5Fu), indicating the activation of the Hh signaling pathway and the correlation between GLI2 and chemoresistance toward 5Fu [158]. GLI2 knockdown decreased ABCG2 expression, and ABCG2 could rescue the effect of GLI2 shRNA in the 5Fu response, verifying that the GLI2-ABCG2 signaling axis is a pivotal mechanism regulating 5Fu resistance in GC cells. Moreover, after the transfection of ZEB7901 siRNA in cisplatin-resistant GC cells, the cell viability was decreased and the cell apoptosis rate was elevated [110]. Additionally, the degree of both was enhanced with increasing concentrations of cisplatin, which indicated that ZEB2 silencing can attenuate the resistance of GC cisplatin.

### 3.9. Other Pathway

Certain unique ZFPs can affect the development of GC via distinct pathways besides the eight routes indicated above. In this part, we mainly focus on C2H2-type ZNF479 and RNF43, which are both upregulated in GC (Figure 3). Jin et al. found that a ZNF479 knockout could inhibit glucose uptake, lactate earnings, adenosine triphosphate amounts and the extracellular acidification rate, leading to cell respiration of oxygen, which is consistent with the high expression of ZNF479 in GC [37]. This study suggested that ZNF479 can regulate glycolysis in part through the β-catenin/c-Myc signaling pathway, which is a master regulator of glycolysis and a key oncogenic driver of tumor proliferation [159,160]. Then, c-Myc can transcriptionally upregulate glycolysis-associated proteins including PKM2 [161], LUT1 [162], LDHA [163] and HK2 [164], thus promoting glucose uptake and the rapid conversion of glucose to lactate. Additionally, mutations in RNF43, a tumor suppressor, were prominently enriched in the H. pylori-induced DNA damage response (DDR) in gastric epithelial cells [165]. Neumeyer et al. discovered that RNF43 governs the DDR in the stomach [97]. A depletion of RNF43 can endow gastric cells with resistance to chemotherapy and radiotherapy and prevent their apoptosis by inhibiting DDR activation. A connection between RNF43 and phosphorylated H2A histone family member X (γH2AX) might be the particular mechanism, suggesting that RNF43 may be the biomarker for therapy selection of GC. Admittedly, with deeper research into the molecular mechanism of GC pathogenesis, there are more and more ZFPs which are recognized to influence GC in unique and unusual ways.

## 4. MZF1: A Double-Edged Sword in GC

Generally, certain kinds of ZFPs only play either carcinogenic or antitumor roles in GC. A growing body of evidence has suggested that myeloid zinc finger 1 (MZF1) constitutes dimers via highly conserved SCAN motifs and may have distinct functions in GC [90]. MZF1 was first identified in humans through research on the hematopoietic cells of the myeloid lineage [166]. The emergence of additional solid tumor cancers, including GC, is increasingly acknowledged to be related to MZF1 as the relevant studies progress. Intriguingly, MZF1 is highly expressed to exert both oncogenic and anticancer effects in GC. In terms of the oncogenic role, the expression of MZF1 was once confirmed to be positively associated with Axl [90], and high Axl expression correlates with GC metastasis [167]. Therefore, it is speculated that MZF1 may bind to and activate the Axl promoter to enhance the expression of Axl, which in turn promotes GC progression, but the possible correlation between MZF1 and Axl in GC warrants further investigation. Fascinatingly, similar to ZNF139 [46] and KLF4 [68], MZF1 can also combine with the promotor of MMPs to function in GC [91]. By integrating with the promoter of MMP-14, MZF1 can facilitate the transcription and expression of MMP-14 to accelerate the EMT process and metastasis of GC cells [91]. Oppositely, MZF1 hinders the proliferation, invasion and metastasis of GC cells. MZF1 can interact with metallothionein 2A (MT2A) to exert an antitumor effect by modulating the transcriptional expression of IκB-α or epigenetically upregulating diallyl trisulfide and docetaxel [88]. In addition, MZF1 upregulates the expression of the tumor suppressor SMAD4, a critical upstream stimulator of TGF-β1 signaling, by enhancing its transcription activity to further inhibit GC progression [87]. Moreover, TGF-β has been extensively recognized as a carcinogen and a tumor suppressor recently [168], similar to MZF1. Fascinatingly, TGF-β can promote tumorigenesis via the mediation of other ZFPs such as GLI1 in GC [76], KLF8 in GC [82] and TWIST1 in colorectal cancer [130]. Additionally, the tumor-related fibroblast phenotype can be induced by osteopontin via the regulation of MZF1 and TGF-β [169]. In cervical cancer, MZF1 can be modulated via TGF-β1-ERK1/2 signaling to obtain the CK17-induced property of cancer stem cells [170]. This evidence proves that TGF-β-mediated MZF1 may play a role in cancer stem cell transformation. Additionally, the overexpression of LODC1 can lead to membrane ectropion of phosphatidylserine (i.e., a landmark event for the early stage of apoptosis), thus curbing tumorigenesis in GC [171]. MZF1 can participate in this process by enhancing LODC1-induced apoptosis and decreasing cell viability, indirectly exerting an inhibitory effect on GC cells [171].

The potential of MZF1 in clinical therapy and the prognosis prediction of GC may not be underestimated. The inhibitory effect of the MZF1-SMAD4 axis may provide new evidence for the discovery of the novel molecular target therapy of GC [87]. Additionally, worse patient prognosis is substantially linked with the downregulation of MT2A and MZF1 [88]. Moreover, many genes encoding ZFPs are the target of miRNA, such as ZNF521 [36], Snail [63] and MZF1. Zheng et al. suggested that the MZF1/miR-328-3p/CD44 pathway might be feasible as a potential candidate for STAD treatment [89] and should be explored. Moreover, miR-337-3p can independently predict GC prognosis via MZF1 and MMP-14 [91]. In conclusion, MZF1 is of importance in metastasis, DNA methylation, apoptosis, prognosis and the possible treatment of GC. MZF1 can facilitate metastasis in GC via the miR-337-3p/MZF1/MMP-14 pathway [91], while impeding the invasion and metastasis of GC cells via the MT2A-NF-κB pathway [88], the MZF1-SMAD4 axis [87] or the MZF1/miR-328-3p/CD44 axis [89]. Additionally, MZF1 can enhance LODC1-induced apoptosis to inhibit GC progression [171].

## 5. ZFPs in Prognosis Prediction and Diagnostic Means

Over the last few years, there has been a remarkable advancement in the identification of novel GC compounds [172,173]. Researchers have identified a multitude of molecular markers (e.g., microsatellite instability, HER2, CDX2 and cell cycle regulators) which may exhibit prognostic potential in GC [174,175,176,177,178]. Several monogenic markers have been utilized to construct prognostic models in GC such as PPARγ [179], HDAC6 [180] and CD44 [181]. Nevertheless, no consensus has been achieved about the optimal biomarkers and methods, despite much work being put into establishing the best tools for prognosis prediction in GC [182,183]. The aforementioned discoveries have summarized numerous ZFPs regulating proliferation, metastasis, drug resistance, DNA methylation and apoptosis in GC cells. Several ZFPs such as ZNF460, ZNF521, Snail, RNF114, ZEB1, ZNF139, ZFP64, GLIS2, GLI1, ZNF545, ZNF471, RNF180, ZIC1, zFOC1, ZBP89 and RNF43 are ascertained to have potential implications for prognosis prediction in GC and can be possibly used to monitor drug efficacy and the recurrence of GC.

Firstly, many ZFPs exert prognostic effects via regulating proliferation, EMT, invasion and metastasis. Given that the ZNF460-APOC1 axis promotes GC progression in vivo [39], ZNF460 can indirectly evaluate the prognosis of GC patients via APOC1 [184]. ZNF521 can also promote proliferation and metastasis via miR-204-5p, thus being highly related to the prognosis of GC individuals [36]. Snail can be maintained by USP13 to trigger EMT and metastasis, resulting in the poor prognosis of GC patients indirectly [58]. RNF114 is negatively associated with the life quality of GC patients by inducing the EMT process and the invasion of GC cells [66]. ZEB1 can activate LAMA4 expression to predict poor overall survival (OS) in GC [107]. In addition, overexpressed ZNF139 can directly affect the prognosis of GC by promoting caspase-3-facilitated apoptosis [47]. In terms of drug resistance, Zhu et al. identified that ZFP64 plays a prognostic role in GC by mediating nab-paclitaxel chemosensitivity [53]. Moreover, the overexpression of both GLIS2 [74] and GLI1 [74,77] is markedly correlated with chemoresistance and worse prognosis in GC.

Secondly, certain ZFPs function as prognostic or diagnostic markers via DNA methylation. In comparison to GC patients with unmethylated or hypomethylated ZNF545 promoters, those with hypermethylated ZNF545 promoters have a substantially shorter median OS [34]. The promoter methylations of ZNF471 [38] and RNF180 [83] are also independent prognostic biomarkers in GC. Fascinatingly, the methylation of RNF180 can be a noninvasive diagnostic target because its low expression can indicate the outset and progression of GC [84,86]. Coincidentally, Chen et al. illustrated that the present criteria for the early diagnosis of GC can be modified by detecting the ZIC1 promoter methylation rate and CEA level [114]. Another two highly expressed ZFPs also exert a diagnostic role in GC. One is zFOC1 which may have the potential to be a tumor biomarker for GC as differentially expressed cDNA [185]. The other is ZBP89 which can bind to the gastrin EGF response element (gERE) to compete with Sp1, thus inhibiting EGF activation [57]. Taniuchi et al. revealed that ZBP89 can be a biomarker for malignant transformation in GC [57].

Pivotally, current research has shown that the RNF43 gene commonly exhibits mutations in GC and RNF43 displays a crucial independent prognostic role in GC [92,93,94,95]. Niu et al. unraveled a tumor suppressor RNF43 and a tight correlation between RNF43, the pathologic tumor–node–metastasis (pTNM) stage and OS [93]. Similarly, Holm et al. illustrated that the median OS was 7.8 months longer in patients with high RNF43 expression compared to their counterparts with low/negative RNF43 expression [92]. In addition, a xenograft model showed that the representative FDA-approved drugs, i.e., docetaxel trihydrate, GSK-2141795 and pelitinib, exhibited obvious inhibitory effects in the RNF43 knockout of GC cells, denoting that these drugs could be novel medications for GC patients with a low expression of RNF43 [94].

## 6. Application of ZFPs in GC Therapeutic Intervention

Recently, preclinical and clinical studies on targeted therapy have also been carried out, and the novel targets researched in GC involve Claudin18.2 (CLDN18.2) [186], FGFR2b monoclonal antibody [187], combinations of immune checkpoint inhibitors and the VEGF receptor-2 (VEGFR-2) antagonist [188]. Due to poor OS, biomarkers have little therapeutic or prognostic value. Fortunately, contemporary research has shown that microRNAs (miRNAs and miRs) are critical regulators of carcinogenesis, which can accurately regulate the expression of downstream ZFPs and further affect the progression of GC (Table 2) [189]. The immense observations of the MicroRNA/ZFPs axis will allow for a new generation of medications to treat advanced cancer by controlling cancer-specific miRs [190].

A large number of miRs are aberrantly expressed in GC, affecting the expression of multiple ZFPs, specifically ZNF521, ZNF139, ZNF24, zinc finger and BTB domain-containing 4 (ZBTB4), Snail, KLF6 and MZF1. The negative link between miR-204-5p and its target gene ZNF521 suggests that downregulating ZNF521 can enhance cell death and inhibit GC proliferation, migration and invasion [36]. miR-195-5p generally decreases in poorly differentiated GC tissues and negatively regulates ZNF139 expression by connecting to the 3’-untranslated area of ZNF139 [191]. miR-940 promotes GC progression by directly binding to the three major untranslated regions of ZNF24 and then downregulating ZNF24 at the post-transcriptional level; so, the targeting of miR-940 may be utilized as a novel strategy for GC therapy in the not-too-distant future [49]. ZBTB4 was a direct target of miR-301b-3p, and their expression in GC was negatively linked [56]. This work is the first to demonstrate that mir-803-301p extensively exists in GC and greatly enhances tumor expansion by restricting ZBTB3 expression. Attractively, the Epstein–Barr virus (EBV) was the first cancer virus discovered to encode microRNA. In EBV-associated gastric cancer (EBVaGC), EBV-encoded BamHI-A rightward transcript microRNAs (EBV-miR-BARTs) arrest the cell cycle in the G2/M phase by targeting Snail, promoting cell apoptosis and inhibiting cell proliferation and migration [63]. KLF6 has been identified as a principal target of miR-181a in GC, and miR-181a may play a carcinogenic role in GC by inhibiting the tumor suppressor KLF6 [69]. In addition, the previously discussed MZF1 fraction suggests the feasibility of the MZF1/miR-328-3p/CD44 axis as a unique, prospective clinical target for STAD. New studies have shown that endogenous miR-337-3p can inhibit GC progression by reducing the MZF1-induced MMP-14 expression pathway. Mechanistically, this is achieved by enrolling Argonaute 2 and triggering repressive chromatin remodeling. Targeting miR-337-3p may also provide a new molecular strategy for GC treatment [91]. Discovering deeper cancer-associated miRs and exploring the clinical application of miRs in GC treatment has become a hot topic that may provide fresh answers and hope for GC patients [193].

It is worth mentioning that other molecules, as important regulators of GC, can target miRs, affect the expression of ZFPs and in turn control the progression of GC, including cyclic RNAs (circRNAs) and long noncoding RNAs (lncRNAs) [194]. circCSNK1G1 can upregulate ZNF217 and then accelerate the progression of GC by promoting the expression of miR-758, implying that circCSNK1G1 could be employed as a diagnostic or therapeutic biomarker for GC [192]. Cui et al. revealed that the lncRNA host gene 20 (lncRNA SNHG20) can prevent GC growth and metastasis by adversely altering miR-495-3p expression, which in turn inhibited ZFX expression [50]. Therefore, the SNHG20/miR-495-3p/ZFX axis could also supply new clues for GC medication. Interestingly, zinc finger antisense 1 (ZFAS1, also known as ZNFX1-AS1), another kind of lncRNA, was recently identified to serve as a proto-oncogene upregulated in GC [195]. The silencing of ZFAS1 can impede the proliferation, EMT and metastatic dissemination of GC cells, along with the attenuated chemoresistance of GC cells toward chemotherapeutics such as cis-platinum or paclitaxel [196]. Besides the aforementioned ZEB2-AS1 [109], the oncogenic feature of ZFAS1 is also dependent on the inhibition of Wnt/β-catenin signaling [196], the miR-200b-3p/Wnt1 axis [197] or the repression of Krüppel-like factor 2 (KLF2) and the NKD inhibitor of the WNT signaling pathway 2 (NKD2) [198]. Thus, the ZFAS1 knockdown can be a therapeutic target with potential clinical significance in GC. Moreover, as circulating IncRNAs can be used to represent the level of circulating tumor cells, an indicator reflecting the malignant progression of GC [199], ZFAS1, which was clarified to be overexpressed and relatively stable in serum, might be a noninvasive diagnostic marker in GC [200,201]. To our knowledge, there is a paucity of research on ZFAS1 in GC; therefore, the clinical potential of this feasible therapeutic approach still warrants further investigation.

## 7. Conclusions

The above provides a brief overview of the research progress on the relationship between ZFPs and GC nowadays, as well as the basic molecular mechanisms as seen through multiple biological processes, including cell proliferation, EMT, invasion and metastasis, inflammation and immune infiltration, apoptosis, cell cycle, DNA methylation, CSCs, drug resistance, etc. Significantly, we highlight the dual inverse role of MZF1 in GC. Despite extensive previous studies in this area, there are still substantial gaps in the complete knowledge of a large number of ZFPs, including ZNF146, ZNF281 and ZDHHC2, wherein their mechanisms of action in GC are still not fully elucidated [202,203].

As a wide-ranging family of eukaryotic transcription variables, ZFPs have exhibited promising potential in several biological fields, such as disease treatment, on account of their specific transcriptional regulation roles [204]. GC remains one of the most prevalent malignancies worldwide, and notwithstanding remarkable breakthroughs in current therapies for GC, such as surgery, chemoradiotherapy and immunotherapy, its prognosis, particularly regarding long-term viability, remains unsatisfactory [205]. Some of the ZFPs identified so far may be oncogenes or tumor suppressors in GC progression, and researchers have designed specific ZFPs to regulate the expression of the corresponding target genes in mammals, which have made great progress [206]. In light of the high intertumoral, intratumoral and interpatient heterogeneity of GC, accurate classification and stratification via reference indicators are indispensable for the diagnosis, treatment and prognosis of GC [207,208,209]. With the in-depth development of bioinformatics for discovering more ZFPs and the elucidation of their molecular mechanisms, staging based on various ZFPs can facilitate more optimized early diagnosis and personalized therapeutics for GC.

## Figures and Tables

**Figure 1 cells-12-01314-f001:**
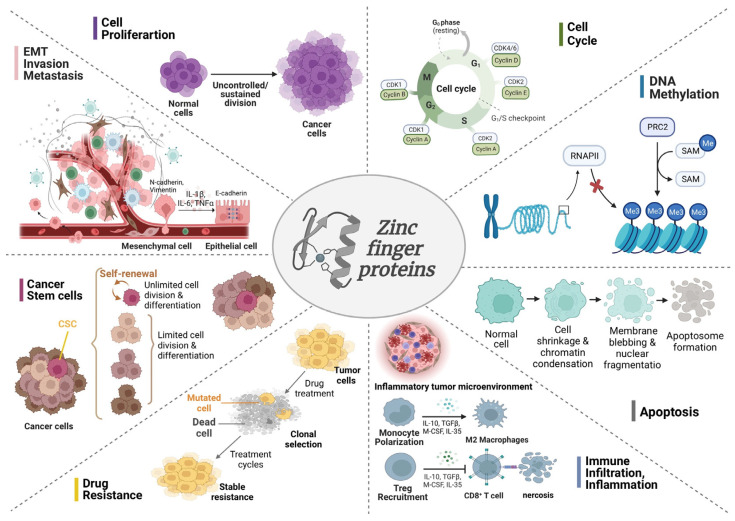
The role of ZFPs in orchestrating the biological events of GC. ZFPs are crucial in the modulation of the eight principal mechanisms in GC cells, involving cell proliferation, EMT, invasion and metastasis, inflammation and immune infiltration, apoptosis, cell cycle, DNA methylation, CSCs and drug resistance.

**Figure 2 cells-12-01314-f002:**
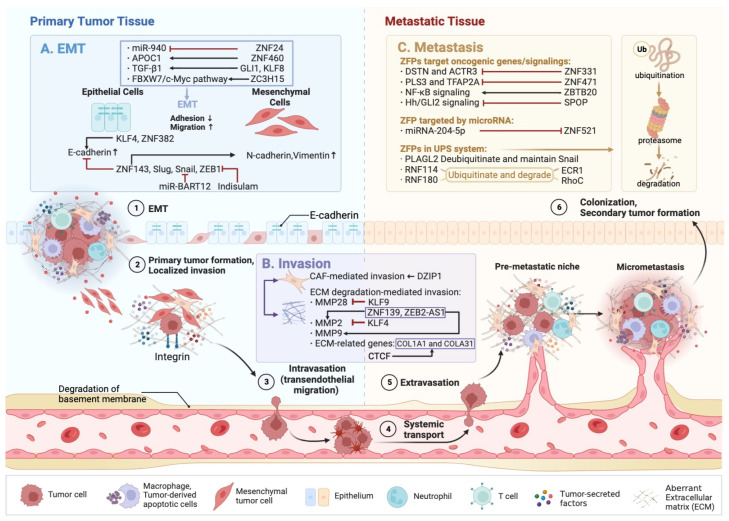
ZFPs regulate EMT, invasion and metastasis. The prerequisite for metastasis and the multi-stage process of metastatic dissemination are as follows: (1) EMT, (2) localized invasion, (3) intravasation, (4) systemic transport via the circulatory system, (5) extravasation and (6) colonization. (**A**) ZFPs including ZNF24, ZNF460, GLI1, KLF8, ZC3H15, KLF4, ZNF382, ZNF143, Slug, Snail and ZEB1 are involved in the EMT process of GC. (**B**) Invasion refers to a process whereby cancer cells invade the adjacent tissues and blood vessels. ZFPs including DZIP1, KLF9, ZNF139 and CTCF, along with zinc finger RNA ZEB2-AS1, participate in CAF-mediated invasion and ECM-mediated invasion. (**C**) Metastasis is another significant hallmark of GC progression, and ZFPs can participate in metastasis in three distinct ways: (1) ZFPs such as ZNF331, ZNF471, ZBTB20 and SPOP can target oncogenic genes or signals. (2) ZFPs such as ZNF521 can be targeted by miRNA-204-5p. (3) ZFPs such as PLAGL2, RNF114 and RNF180 are involved in the UPS system to exert tumor-suppressive or pro-tumorigenic effects.

**Figure 3 cells-12-01314-f003:**
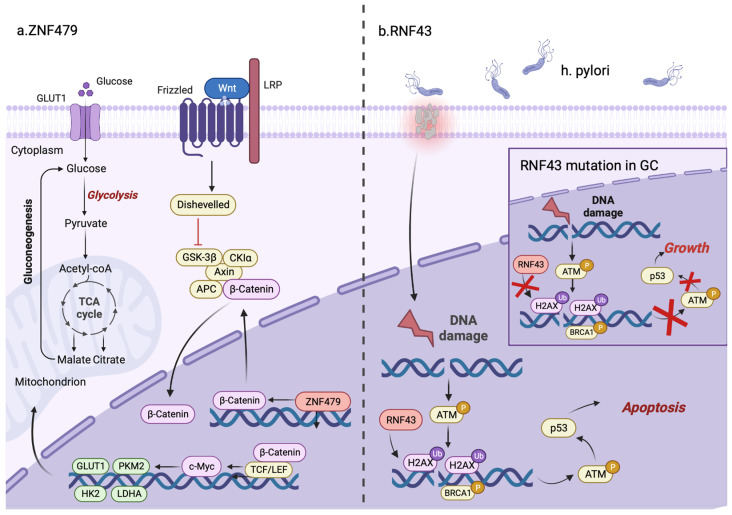
ZNF479 and RNF43 exert their effects via unique signaling pathways in GC. (**a**) ZNF479 transcriptionally activates the target gene β-catenin that binds to the TCF/LEF transcription factor and triggers the expression of c-Myc. Furthermore, c-Myc stimulates downstream target genes containing GLUT1, PKM2, HK2 and LDHA, thereby promoting glycolysis and cell survival in GC. (**b**) RNF43 mutations are prevalent in the H. pylori-induced DNA damage response (DDR) in gastric epithelial cells. Typically, RNF43 induces apoptosis in cells with DDR by directly interacting with and ubiquitinating H2AX. However, when RNF43 is mutated, the loss-of-function can decrease the level of H2AX ubiquitination and reverse DDR-induced apoptosis, thus enhancing cell growth in GC.

**Table 1 cells-12-01314-t001:** Representative ZFPs and the corresponding signaling pathways in GC.

ZFPs	Aliases	Expression	Main Roles	Targets	Ref
ZNF852	-	↑	Cell proliferation Drug resistance Cancer stem cells	EGFR	[33]
ZNF545	KIAA1948, MGC45380, ZFP82	↓	Cell proliferation DNA methylation	rRNA, Heterochromatin protein 1β, Trimethylated histone H3	[34,35]
ZNF521	EHZF, Evi3	↑	Cell proliferation EMT, invasion and metastasis	MicroRNA-106-5p	[36]
ZNF479	HKr19	↑	Cell proliferation Glycolysis	β-catenin/c-Myc pathway	[37]
ZNF471	KIAA1396, Z1971, Zfp78	↓	Cell proliferation EMT, invasion and metastasis DNA methylation	KAP1, TFAP2A, PLS3	[38]
ZNF460	IKZF2, Helios, ZNFN1A2	↑	Cell proliferation EMT, invasion and metastasis Cell cycle	APOC1	[39]
ZNF331	RITA, ZNF361, ZNF463	↓	Cell proliferation EMT, invasion and metastasis DNA methylation	DSTN, EIF5A, GARS, DDX5, STAM, UQCRFS1, SET, DSTN, ACTR3, SSBP1, PNPT1	[21,40]
ZNF280B	5′OY11.1, SUHW2, ZNF279, ZNF632	↑	Cell proliferation	---	[41]
ZNF143	pHZ-1, SBF, STAF	↑	Cell proliferation Apoptosis EMT, invasion and metastasis	ROS/p53 axis, PI3K/Akt pathway	[42,43]
ZNF139	ZNF36, ZSCAN33, ZKSCAN1, KOX18, PHZ-37	↑	Cell proliferation EMT, invasion and metastasis Drug resistance Apoptosis Cell cycle	Bcl-2, Bax, Caspase-3, MDR1/P-gp, MRP-2, Bcl-185, ANXA2, Fascin, PDXK, MMP-2, MMP-9, ICAM-1, TIMP-1	[44,45,46,47,48]
ZNF24	KOX17, ZFP191, ZNF191, ZSCAN3	↓	EMT, invasion and metastasis	MicroRNA-940	[29,49]
ZFX	ZNF926	↑	Cell proliferation Apoptosis Cell cycle	ERK-MAPK pathway, SNHG20/miR-495-3p/ZFX axis, FTX/miR-144/ZFX axis	[50,51]
ZHX2	KIAA0854	↑	Inflammation		[52]
ZFP64	ZNF338, dJ548G19.1dJ831D17.1FLJ10734 FLJ12628	↑	EMT, invasion and metastasis Drug resistance Cancer stem cells Immunosuppress	GAL-1	[53]
ZC3H15	LEREPO4	↑	Cell proliferation EMT, invasion and metastasis	FBXW7, c-Myc, FBXW7/c-Myc pathway	[54]
ZBTB20	DKFZp566F123, DPZF, ODA-8S, ZNF288	↑	Cell proliferation EMT, invasion and metastasis	NF-κBp65, MMP-2, MMP-9, IκBα	[55]
ZBTB4	KAISO-L1, KIAA1538, ZNF903	↓	Cell proliferation	miR-301b-3p	[56]
ZBP89	BERF-1, BFCOL1, HT-BETA, pHZ-52, ZFP148, ZNF148	↑	Cell proliferation	gERE, Sp1, EGF	[57]
TWIST1	ACS3, bHLHa38, BPES2, BPES3, CRS, CRS1, H-twist, SCS, TWIST		EMT, invasion and metastasis	E-cadherin, Snail, Zeb and Twist	[58]
SPOP	BTBD32, TEF2	↓	Cell proliferation EMT, invasion and metastasis Apoptosis	Hh/GLI2 pathway	[59,60]
Snail	SNAI1, SNAIL1, SLUGH2, SNA, SNAH	↑	EMT, invasion and metastasis	USP13, EBV-miR-BART12, E-cadherin, TNF-α, NF-κB pathway	[61,62,63,64]
DZIP1	DZIP, KIAA0996	↑	Cell proliferation EMT, invasion and metastasis Immunosuppress	CAFs	[65]
Slug	SNAI2, SNAIL2, SLUH1, SLUGH,	↑	EMT, invasion and metastasis	E-cadherin, SIP1, SIP2, Snail	[61]
RNF114	PSORS12, ZNF313	↑	Cell proliferation EMT, invasion and metastasis	EGR1, miR-218-5p, EGR1	[66]
PLAGL2	ZNF900	↑	Cell proliferation EMT, invasion and metastasis	USP37, Snail	[62]
MORC2	AC004542.C22.1, KIAA0852, ZCW3, ZCWCC1	↑	Cell proliferation EMT, invasion and metastasis Cell cycle	C/EBPα	[67]
KLF4	EZF, GKLF	↓	Cell proliferation EMT, invasion and metastasis	β-catenin, E-cadherin, MMP2	[68]
KLF6	BCD1, COPEB, CPBP, GBFPAC1, ST12, Zf9	↓	Cell proliferation Cell cycle	p21, c-Myc, LINC00703, miR-181a/KLF6 axis	[69,70]
KLF9	BTEB1	↓	EMT, invasion and metastasis	MMP28, TPTEP1/KLF9/PER1 axis, miR-548d-3p	[71,72]
GLIS2	NPHP7	↑	Drug resistance Cell cycle	Cyclin D1, 𝛽-catenin, TCF/LEF	[73,74,75]
GLI1	GLI	↑	EMT, invasion and metastasis Drug resistance Apoptosis Cancer stem cell	E-cadherin, Vimentin, TGF-β1, GANT, PI3K/Akt/mTOR pathway, PD-L1, Akt-mTOR-p1S70K, HER2, SMO	[76,77,78]
GLI2	-	↑	Cell cycle Cancer stem cell Drug resistance	GANT61, Hh/Gli pathway, CyclinD1, p21, Sp1, hTERT/Sp1/Gli1 axis	[59]
CTCF	CFAP108, FAP108	↑	Cell proliferation EMT, invasion and metastasis	Wnt pathway, COL1A1, COLA31	[79]
TNFAIP3	A20, OTUD7C	↓	Apoptosis Inflammation	NF-κB pathway, TNF-α, IL-1, IL-1, IL-6, IL-8, CARMA1-Bcl-10-MALT1 pathway, TNFR1, TNFR2	[80,81]
KLF8		↑	EMT, invasion and metastasis	E-cadherin, Vimentin, TGF-β1	[82]
RNF180	-	↓	DNA methylation EMT, invasion and metastasis	ZIC2	[83,84,85,86]
MZF1	MZF-1, MZF1B, Zfp98, ZNF42, ZSCAN6	↓	Apoptosis EMT, invasion and metastasis	MT2A, NF-κB pathway, LODC1, SMAD4, miRNA-337-3p, MMP-14	[87,88,89,90,91]
RNF43	DKFZp781H0392, FLJ20315, URCC	↓	Apoptosis EMT, invasion and metastasis Cancer stem cell	Wnt signaling pathway	[13,92,93,94,95,96,97,98]
RNF2	BAP-1, BAP1, DING, HIPI3, RING1B, RING2	↑	Cell cycle Cell proliferation	RASSF10/NPM/RNF2 feedback	[99,100]
ZNF703	FLJ14299, NLZ1, ZEPPO1, ZNF503L, Zpo1	↑	Cell proliferation	LBX2-AS1, NFIC, miR-491-5p	[101]
KLF12	AP-2rep, AP2REP, HSPC122	↑	Cell proliferation Cell cycle	AP-2-alpha gene, A32, miR-137	[102,103]
MPS-1	RPS27, S27	↑	EMT, invasion and metastasis Apoptosis	MPS-1/NF-κB/Gadd45β pathway, JNK, Caspase3	[104,105,106]
ZEB1	AREB6, BZP, FECD6, NIL-2-A, PPCD3, TCF8, ZEB, Zfhep, Zfhx1a	↑	EMT, invasion and metastasis	LAMA4, MMP2, indisulam, RBM39, DCAF15, Yes, miR-200a, miR200b, miR-200c, miR-141, miR-429, N-cadherin, BZLF1	[58,80,107,108]
ZEB2	KIAA0569, SIP-1, SIP1, ZFHX1B	↑	Cell proliferation EMT, invasion and metastasis Drug resistance	MMP-2, MMP-9	[109,110]
ZNF382	FLJ14686, KS1	↓	EMT, invasion and metastasis DNA methylation Cancer stem cell	SNAIL, Vimentin, Twist, NOTCH1, NOTCH2, NOTCH3, NOTCH4, HES-1, JAG1, MMP2, MMP11, NANOG, OCT4, SOX2, E-cadherin	[111]
ZNRF3	BK747E2.3, FLJ22057, KIAA1133, RNF203	↓	Apoptosis Cell proliferation	β-catenin, TCF-4, Wnt/β-catenin/TCF pathway	[112,113]
ZIC1	ZIC, ZNF201	↓	EMT, invasion and metastasis DNA methylation Cell cycle	Shh signaling, PI3K/Akt signaling, MAPK/ERK signaling	[114,115]

↓ indicates that the protein is down-regulated in GC. ↑ indicates that the protein is up-regulated in GC.

**Table 2 cells-12-01314-t002:** The typical interaction between microRNAs and ZFPs in GC.

miRs	Expression	Targets	Functions	Ref
miR-204-5p	↓	ZNF521	Negatively regulates ZNF521 Cell proliferation EMT, invasion and metastasis Apoptosis	[36]
miR-195-5p	↓	ZNF139	Negatively regulates ZNF139 Cell proliferation Drug resistance	[191]
miR-940	↑	ZNF24	Negatively regulates ZNF24 Cell proliferation EMT, invasion and metastasis	[49]
miR-301b-3p	↑	ZBTB4	Negatively regulates ZBTB4 Cell proliferation Apoptosis Cell cycle	[56]
EBV-miR-BARTs	↓	Snail	Negatively regulates Snail Cell proliferation EMT, invasion and metastasis Apoptosis Cell cycle	[63]
miR-181a	↑	KLF6	Negatively regulates KLF6 Cell proliferation EMT, invasion and metastasis	[69]
miR-337-3p	↓	MZF1	Negatively regulates MZF1 Cell proliferation EMT, invasion and metastasis	[91]
miR-758	↓	ZNF217	circCSNK1G1 negatively regulates miR-758, while miR-758 negatively regulates ZNF217 Cell proliferation	[192]
miR-495-3p	↓	ZFX	lncRNA SNHG20 negatively regulates miR-495-3p, while miR-495-3p negatively regulates ZFX Cell proliferation (SNHG20/miR-495-3p/ZFX axis) EMT, invasion and metastasis	[50]

↓ indicates that the miR is down-regulated in GC. ↑ indicates that the miR is up-regulated in GC.

## Data Availability

All data included in this study are available upon request by contacting the corresponding author.

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
