# Peer review of "Zinc Finger Proteins in the War on Gastric Cancer: Molecular Mechanism and Clinical Potential"

_cells, 2023, doi:10.3390/cells12091314_

Round 1

Reviewer 1 Report

The manuscript is well written and interesting. I therefore suggest this manuscript to be published after minor revisions. I would suggest the authors to consider following:

l   I recommend to extend the section 3.7 “ZFPs Regulate Cancer Stem Cells” with more literature and more in detail description of ZFPs in CSCs. For example, CD44 is an important surface marker of gastric cancer stem cells. The study reported that enhanced miR-144-3p expression inhibited cancer stemness and CD44 expression of gastric CSCs by targeting Gli2 (Lu et al., 2021). Moreover, downregulation of Gli1 inhibited CD44 and cancer stemness in gastric cancer cells was also demonstrated in previous study (Yu et al., 2014). I recommend to cite these references in the manuscript (in table 1, 2 and section 3.7).

Yu D, Shin HS, Lee YS, Lee D, Kim S, Lee YC. Genistein attenuates cancer stem cell characteristics in gastric cancer through the downregulation of Gli1. Oncol Rep. 2014;31(2):673-8.

Lu Y, Zhang B, Wang B, Wu D, Wang C, Gao Y, Liang W, Xi H, Wang X, Chen L. MiR-144-3p inhibits gastric cancer progression and stemness via directly targeting GLI2 involved in hedgehog pathway. J Transl Med. 2021;19(1):432.

l   Tang et al., indicated that NOX4 regulates gastric cancer cell proliferation or apoptosis through Gli1 pathway. I would recommend to discuss the role of NOX4 and Gli1 in the regulation of gastric cancer cell apoptosis in section 3.4. ZFPs Regulate Apoptosis.

Tang CT, Lin XL, Wu S, Liang Q, Yang L, Gao YJ, Ge ZZ. NOX4-driven ROS formation regulates proliferation and apoptosis of gastric cancer cells through the GLI1 pathway. Cell Signal. 2018;46:52-63.

l   ABCG2 is known to contribute to multidrug resistance in gastric cancer chemotherapy. For example, ABCG2 is reported to be an important mediator for Gli2-associated 5Fu resistance (Yu et al., 2017). I recommend to discuss the role of ABCG2 and ZFPs in the regulation of gastric cancer cell drug resistance in section 3.8.

Yu B, Gu D, Zhang X, Liu B, Xie J. The role of GLI2-ABCG2 signaling axis for 5Fu resistance in gastric cancer. J Genet Genomics. 2017;44(8):375-383.

General Comments:

l   Line 14. “Zn2+” should be “Zn2+”.

l   Line 30.  “TNM staging”….I would suggest to show the full name instead of its abbreviation. “TNM (Tumor, Node, Metastasis) staging….”.

l   Line 71. “Zn2+” should be “Zn2+”.

l   Line 89. “Zn2+” should be “Zn2+

l   Line 199. “thuattenuatingte the migration….” Should be “thus attenuating”.

l   Line 204. “in vitro” should be “in vitro”.

l   Line 223. “through TGF-b1” should be “through transforming growth factor b1 (TGF-b1)”.

l   Line 261. “in vitro and in vivo” should be “in vitro and in vivo”.

l   Line 264. “in vitro” should be “in vitro

l   Line 280. “CD8+ T cells” should by “CD8+ T cells”

l   Line 303. “in vivo” should be “in vivo”.

l   Line 331. “p65 and NF-B” should be “p65 and NF-kB”.

l   Line 333. “NF-B target” should be “NF-kB target”.

l   Line 500. “of transforming growth factor beta1 (TGF-β) signaling” should be “of TGF-b1 signaling”.

l   Line 585 “microRNAs” should be “miRNAs”.

l   Table 2 “microRNA-204-5p” should be “miR-204-5p”.

l   Line 600 “miRNA-940 (miRNA 940) may…” should be “miR-940 may…”.

Moderate editing of English language is required.

Author Response

Response to Reviewer 1 Comments

Point 1: I recommend to extend the section 3.7 “ZFPs Regulate Cancer Stem Cells” with more literature and more in detail description of ZFPs in CSCs. For example, CD44 is an important surface marker of gastric cancer stem cells. The study reported that enhanced miR-144-3p expression inhibited cancer stemness and CD44 expression of gastric CSCs by targeting Gli2 (Lu et al., 2021). Moreover, downregulation of Gli1 inhibited CD44 and cancer stemness in gastric cancer cells was also demonstrated in previous study (Yu et al., 2014). I recommend to cite these references in the manuscript.

Response 1: Thank you very much for giving such detailed writing guidance! Based on your suggestions and references, we have added more literature and detailed descriptions of ZFPs in CSCs. (Line 1275-1374 in the track pdf version of revised manuscript)

The nuclear transcription factors GLI1 and GLI2 are essential molecules in the Shh signaling pathway, which have been implicated in sustaining CSC properties in GC [150]. Studies have revealed that GLI1 not only participates in the tumorigenesis of GC, but also upregulates CSC surface markers like CD44, Lysine-specific demethylase 1 (LSD1) and SRY-Box Transcription Factor 9 (Sox9) [151,152], whereas GLI2 fosters the expression of CSC-related genes, such as CD44, Nanog homeobox (Nanog) and octam-er-binding transcription factor 4 (Oct4), by inducing Platelet derived growth factor re-ceptor beta (PDGFRB) [153,154]

Point 2: Tang et al., indicated that NOX4 regulates gastric cancer cell proliferation or apoptosis through Gli1 pathway. I would recommend to discuss the role of NOX4 and Gli1 in the regulation of gastric cancer cell apoptosis in section 3.4. ZFPs Regulate Apoptosis.

Response 2: Thank you for giving us this suggestion. We have added the role of NOX4 and Gli1 in the regulation of gastric cancer cell apoptosis in section 3.4. ZFPs Regulate Apoptosis based on the references you provided. (Line 820-1041 in the track pdf version of revised manuscript)

GLI1 knockdown can reverse the effects of NOX4 overexpression, demonstrating that GLI1 was an indispensable effector of NOX4 in mediating cell apoptosis [137]. Moreo-ver, GLI1 was significantly decreased in the NOX4 knockdown group, accompanied by a reduction of apoptotic proteins such as Bcl2, Bax and cleaved PARP, confirming that NOX4 might stimulate GLI1 and promote apoptotic proteins expression to facilitate apoptosis. Moreover, itraconazole, an effective therapeutic drug based on GLI1 regula-tion of downstream targets Bax and PARP, also effectively demonstrated the close re-lationship between GLI1 and apoptosis [138]

Point 3: ABCG2 is known to contribute to multidrug resistance in gastric cancer chemotherapy. For example, ABCG2 is reported to be an important mediator for Gli2-associated 5Fu resistance (Yu et al., 2017). I recommend to discuss the role of ABCG2 and ZFPs in the regulation of gastric cancer cell drug resistance in section 3.8 ZFPs Regulate Drug Resistance.

Response 3: Thank you for giving us this suggestion. In accordance with your suggestion and references, we have added the role of ABCG2 and ZFPs in the regulation of gastric cancer cell drug resistance in section 3.8. ZFPs Regulate Drug Resistance. (Line 1412-1417 in the track pdf version of revised manuscript)

Beiqin Yu et al. discovered that GLI2 expression was elevated in GC cells treated with fluorouracil (5FU), indicating the activation of Hh signaling pathway and the correlation between GLI2 and chemoresistance towards 5Fu [158]. GLI2 knockdown de-creased ABCG2 expression, and ABCG2 could rescue the effect of GLI2 shRNA in 5Fu response, verifying that the GLI2-ABCG2 signaling axis is a pivotal mechanism regulating 5Fu resistance in GC cells.

Point 4: General Comments: “Zn2+” should be “Zn2+”; “TNM staging….” I would suggest to show the full name instead of its abbreviation; “thuattenuatingte the migration…” Should be “thus attenuating”; “in vitro” should be “in vitro”; “through TGF-b1” should be “through transforming growth factor b1 (TGF-b1)”; “CD8+ T cells” should by “CD8+ T cells”; “in vivo” should be “in vivo”; “NF-B target” should be “NF-kB target”; Table 2 “microRNA-204-5p” should be “miR-204-5p” and so on.

Response 4: Thank you very much for giving our work a generally favorable evaluation,and it is so kind of you to provide us with such detailed improvements. We apologize for our negligence. The corrections have been made accordingly. As your suggestion, the correct form of “Zn2+ has been applied. (Line 14, Line 247, Line 298)

We have used the full name (Tumor, Node, Metastasis) of TNM. (Line 39)

The correct forms of “in vitro” and “in vivo” have been applied. (Line 696, Line 698, Line 729, Line 776, Line 779, Line 813, Line 1815)

We have revised “thuattenuatingte the migration...” to “thus attenuating...”. (Line 693)

The errors in full name and abbreviation of TGF-b1 have been corrected. (Line 725)

The correct form of “CD8+ T cells” has been applied. (Line 796)

We have revised “NF-B” to “NF-kB”. (Line 1039)

Based on your suggestion, we have standardized the abbreviations of “microRNA”. (Line 1942)

Reviewer 2 Report

In this report the authors provide an overview of the role of zinc finger proteins in gastric cancer. They discuss the role of these proteins in the regulation of cell proliferation, EMT, metastasis, inflammation, apoptosis, and drug resistance in relation to various signaling pathways, diagnosis and potential therapy. Overall the manuscript provide a good overview and will be useful summary for investigators inside and outside the ZFP field. Overall the paper is well written.

1.     Line71: “fold” maybe better to write the “ structure” instead.

2.     Review could benefit from a schematic illustrating the effect of various ZFPs on EMT and migration/invasiveness.

3.     Line 548: “[59].TNF114.  Please add spacing.

4.     With respect to potential therapy: is administering anti-sense RNAs targeting the expression of ZFPs that stimulate the oncogenic phenotype a potential alternative therapeutic approach? Any reports on this?

Overall the paper is well written.

Author Response

Response to Reviewer 2 Comments

Point 1: Line71: “fold” maybe better to write the “structure” instead.

Response 1: Thank you for pointing it out. In accordance with your suggestion, we have revised “fold” to “structure”. (Line 247 in the track pdf version of revised manuscript)

Point 2: Review could benefit from a schematic illustrating the effect of various ZFPs on EMT and migration/invasiveness.

Response 2: Thank you for your opinion which we quite agree with. We have added a schematic illustrating (Figure 2) and the corresponding figure legend in 3.2. ZFPs Regulate EMT, Invasion and Metastasis. By regrouping the molecules of different ZFPs in three different pathways (EMT, invasion and metastasis), we sincerely hope this illustration can make our statement more explicit and vivid. (Line 529-683 in the track pdf version of revised manuscript)

Figure 2. ZFPs regulate EMT, invasion and metastasis. The prerequisite for metastasis and the multi-stage process of metastatic dissemination are: 1) EMT, 2) localized invasion, 3) intravasation, 4) systemic transport via the circulatory system, 5) extravasation, and 6) colonization. (A) ZFPs including ZNF24, ZNF460, GLI1, KLF8, ZC3H15, KLF4, ZNF382, ZNF143, Slug, Snail and ZEB1 are involved in EMT process of GC. (B) Invasion refers to a process in that cancer cells invade the adjacent tissues and blood vessels. ZFPs including DZIP1, KLF9, ZNF139, CTCF, along with zinc finger RNA ZEB2-AS1, participate in CAF-mediated invasion and ECM-mediated invasion. (C) Metastasis is another significant hallmark of GC progression, and ZFPs can participate in metastasis via three distinct ways: 1) ZFPs such as ZNF331, ZNF471, ZBTB20 and SPOP can target oncogenic genes or signalings. 2) ZFPs such as ZNF521 can be targeted by miRNA-204-5p. 3) ZFPs such as PLAGL2, RNF114, and RNF180 are involved in the UPS system to exert tumor-suppressive or pro-tumorigenic effects.

Point 3: Line 548: “[59].TNF114.” Please add spacing.

Response 3: Thank you for your reminder. We apologize for our negligence. We have added the spacing. (Line 1820 in the track pdf version of revised manuscript)

Point 4: With respect to potential therapy: is administering anti-sense RNAs targeting the expression of ZFPs that stimulate the oncogenic phenotype a potential alternative therapeutic approach? Any reports on this?

Response 4: We are very grateful to you for this comment. After reviewing and searching the published articles, we found that anti-sense RNAs can indeed be used in cancer treatment, and it is a very novel and promising research area. Fortunately, in the case of gastric cancer, some other relevant research is already published besides the report on ZEB2-antisense RNA1 (ZEB2-AS1) mentioned before. We have added some contents to 3.2. ZFPs Regulate EMT, Invasion and Metastasis. Considering that ZEB2-AS1 belongs to long non-coding RNA, corresponding content is also added in 6. Application of ZFPs in GC Therapeutic Intervention. Although none of these studies are newly published, it still reminds us of the clinical potential of this feasible therapeutic approach, and we genuinely hope that this revision provides a good answer to this comment. (Line 2004-2018 in the track pdf version of revised manuscript)

Interestingly, Zinc finger antisense 1 (ZFAS1, a.k.a. ZNFX1-AS1), another kind of lncRNAs, was currently identified to serve as a proto-oncogene upregulated in GC [195]. The silencing on ZFAS1 can impede the proliferation, EMT, and metastatic dis-semination of GC cells, along with the attenuated chemoresistance of GC cells towards chemotherapeutics such as cis-platinum or paclitaxel [196]. Besides the aforementioned ZEB2-AS1 [109], the oncogenic feature of ZFAS1 is also dependent on the inhi-bition of Wnt/β-catenin signaling [196], the miR-200b-3p/Wnt1 axis [197] or the repression of Krüppel-like factor 2 (KLF2) and NKD inhibitor of WNT signaling pathway 2 (NKD2) [198]. Thus, ZFAS1 knockdown can be a therapeutic target with potential clinical significance in GC. Moreover, as circulating lncRNAs can be used to represent the level of circulating tumor cells, an indicator reflecting the malignant progression of GC [199], ZFAS1 which was clarified to be overexpressed and relatively stable in serum might be a noninvasive diagnostic marker in GC [200,201]. To our knowledge, there is a paucity of the research of ZFAS1 in GC, therefore, the clinical potential of this feasible therapeutic approach still warrants further investigation.
